# CROSS-LINGUAL ALIGNMENT VS JOINT TRAINING: A COMPARATIVE STUDY AND A SIMPLE UNIFIED FRAMEWORK

**Zirui Wang**[*]**, Jiateng Xie**[*]**, Ruochen Xu, Yiming Yang, Graham Neubig, Jaime Carbonell**
Language Technologies Institute, Carnegie Mellon University, Pittsburgh, PA 15213, USA
`{ziruiw, jiatengx, ruochenx, yiming, gneubig, jgc}@cs.cmu.edu`

## ABSTRACT

Learning multilingual representations of text has proven a successful method for many cross-lingual transfer learning tasks. There are two main paradigms for learning such representations: (1) alignment, which maps different independently trained monolingual representations into a shared space, and (2) joint training, which directly learns unified multilingual representations using monolingual and cross-lingual objectives jointly. In this paper, we first conduct direct comparisons of representations learned using both of these methods across diverse cross-lingual tasks. Our empirical results reveal a set of pros and cons for both methods, and show that the relative performance of alignment versus joint training is task-dependent. Stemming from this analysis, we propose a simple and novel framework that combines these two previously mutually-exclusive approaches. Extensive experiments demonstrate that our proposed framework alleviates limitations of both approaches, and outperforms existing methods on the MUSE bilingual lexicon induction (BLI) benchmark. We further show that this framework can generalize to contextualized representations such as Multilingual BERT, and produces state-of-the-art results on the CoNLL cross-lingual NER benchmark.[1]

## 1 INTRODUCTION

Continuous word representations (Mikolov et al., 2013a; Pennington et al., 2014; Bojanowski et al., 2017) have become ubiquitous across a wide range of NLP tasks. In particular, methods for *cross-lingual word embeddings* (CLWE) have proven a powerful tool for cross-lingual transfer for downstream tasks, such as text classification (Klementiev et al., 2012), dependency parsing (Ahmad et al., 2019), named entity recognition (NER) (Xie et al., 2018; Chen et al., 2019), natural language inference (Conneau et al., 2018b), language modeling (Adams et al., 2017), and machine translation (MT) (Zou et al., 2013; Lample et al., 2018a; Artetxe et al., 2018b; Lample et al., 2018b). The goal of these CLWE methods is to learn embeddings in a *shared vector space* for two or more languages. There are two main paradigms for learning CLWE: *cross-lingual alignment* and *joint training*.

The most successful approach has been the cross-lingual embedding alignment method (Mikolov et al., 2013b), which relies on the assumption that monolingually-trained continuous word embedding spaces share similar structure across different languages. The underlying idea is to first independently train embeddings in different languages using monolingual corpora alone, and then learn a mapping to align them to a shared vector space. Such a mapping can be trained in a supervised fashion using parallel resources such as bilingual lexicons (Xing et al., 2015; Smith et al., 2017; Joulin et al., 2018b; Jawanpuria et al., 2019), or even in an unsupervised[2] manner based on distribution matching (Zhang et al., 2017a; Conneau et al., 2018a; Artetxe et al., 2018a; Zhou et al., 2019). Recently, it has been shown that alignment methods can also be effectively applied to contextualized word representations (Schuster et al., 2019; Aldarmaki & Diab, 2019).

---

[*]Equal contribution. Ordering determined by dice rolling.

[1]As of the time of submission Sept 25, 2019. Source code is available at `https://github.com/thespectrewithin/joint-align`.

[2]In this paper, "supervision" refers to that provided by a parallel corpus or bilingual dictionaries.

Another successful line of research for CLWE considers joint training methods, which optimize a monolingual objective predicting the context of a word in a monolingual corpus along with either a hard or soft cross-lingual constraint. Similar to alignment methods, some early works rely on bilingual dictionaries (Ammar et al., 2016; Duong et al., 2016) or parallel corpora (Luong et al., 2015; Gouws et al., 2015) for direct supervision. More recently, a seemingly naive *unsupervised* joint training approach has received growing attention due to its simplicity and effectiveness. In particular, Lample et al. (2018b) reports that simply training embeddings on concatenated monolingual corpora of two related languages using a shared vocabulary without any cross-lingual resources is able to produce higher accuracy than the more sophisticated alignment methods on unsupervised MT tasks. Besides, for contextualized representations, unsupervised multilingual language model pretraining using a shared vocabulary has produced state-of-the-art results on multiple benchmarks[3] (Devlin et al., 2019; Artetxe & Schwenk, 2019; Lample & Conneau, 2019).

Despite a large amount of research on both alignment and joint training, previous work has neither performed a systematic comparison between the two, analyzed their pros and cons, nor elucidated when we may prefer one method over the other. Particularly, it's natural to ask: (1) Does the phenomenon reported in Lample et al. (2018b) extend to other cross-lingual tasks? (2) Can we employ alignment methods to further improve unsupervised joint training? (3) If so, how would such a framework compare to supervised joint training methods that exploit equivalent resources, i.e., bilingual dictionaries? (4) And lastly, can this framework generalize to contextualized representations?

In this work, we attempt to address these questions. Specifically, we first evaluate and compare alignment versus joint training methods across three diverse tasks: BLI, cross-lingual NER, and unsupervised MT. We seek to characterize the conditions under which one approach outperforms the other, and glean insight on the reasons behind these differences. Based on our analysis, we further propose a simple, novel, and highly generic framework that uses unsupervised joint training as initialization and alignment as refinement to combine both paradigms. Our experiments demonstrate that our framework improves over both alignment and joint training baselines, and outperforms existing methods on the MUSE BLI benchmark. Moreover, we show that our framework can generalize to contextualized representations such as Multilingual BERT, producing state-of-the-art results on the CoNLL cross-lingual NER benchmark. To the best of our knowledge, this is the first framework that combines previously mutually-exclusive alignment and joint training methods.

## 2 BACKGROUND: CROSS-LINGUAL REPRESENTATIONS

**Notation.** We assume we have two different languages $\{L_1, L_2\}$ and access to their corresponding training corpora. We use $V_{L_i} = \{w_{L_i}^j\}_{j=1}^{n_{L_i}}$ to denote the vocabulary set of the $i$th language where each $w_{L_i}^j$ represents a unique token, such as a word or subword. The goal is to learn a set of embeddings $E = \{x^j\}_{j=1}^m$, with $x^j \in \mathbb{R}^d$, in a *shared* vector space, where each token $w_{L_i}^j$ is mapped to a vector in $E$. Ideally, these vectorial representations should have similar values for tokens with similar meanings or syntactic properties, so they can better facilitate cross-lingual transfer.

### 2.1 ALIGNMENT METHODS

Given the notation, alignment methods consist of the following steps:
**Step 1:** Train an embedding set $E_0 = E_{L_1} \cup E_{L_2}$, where each subset $E_{L_i} = \{x_{L_i}^j\}_{j=1}^{n_{L_i}}$ is trained independently using the $i$th language corpus and contains an embedding $x_{L_i}^j$ for each token $w_{L_i}^j$.
**Step 2:** Obtain a seed dictionary $D = \{(w_{L_1}^i, w_{L_2}^j)\}_{k=1}^K$, either provided or learnt unsupervised.
**Step 3:** Learn a projection matrix $W \in \mathbb{R}^{d \times d}$ based on $D$, resulting in a final embedding set $E_A = (W \cdot E_{L_1}) \cup E_{L_2}$ in a shared vector space.

To find the optimal projection matrix $W$, Mikolov et al. (2013b) proposed to solve the following optimization problem:

$$\min_{W \in \mathbb{R}^{d \times d}} \|W X_{L_1} - X_{L_2}\|_F \tag{1}$$

---

[3] https://github.com/google-research/bert/blob/master/multilingual.md

where $X_{L_1}$ and $X_{L_2}$ are matrices of size $d \times K$ containing embeddings of the words in $D$. Xing et al. (2015) later showed further improvement could be achieved by restricting $W$ to an orthogonal matrix, which turns the Eq.(1) into the Procrustes problem with the following closed form solution:

$$W^* = UV^T, \tag{2}$$

$$\text{with } U\Sigma V^T = \text{SVD}(X_{L_2}X_{L_1}^T) \tag{3}$$

where $W^*$ denotes the optimal solution and $\text{SVD}(\cdot)$ stands for the singular value decomposition.

As surveyed in Section 5, different methods (Smith et al., 2017; Conneau et al., 2018a; Joulin et al., 2018b; Artetxe et al., 2018a) differ in the way how they obtain the dictionary $D$ and how they solve for $W$ in step 3. However, most of them still involve solving the Eq.(2) as a crucial step.

## 2.2 Joint Training Methods

Joint training methods in general have the following objective:

$$\mathcal{L}_J = \mathcal{L}_1 + \mathcal{L}_2 + \mathcal{R}(L_1, L_2) \tag{4}$$

where $\mathcal{L}_1$ and $\mathcal{L}_2$ are monolingual objectives and $\mathcal{R}(L_1, L_2)$ is a cross-lingual regularization term. For example, Klementiev et al. (2012) use language modeling objectives for $\mathcal{L}_1$ and $\mathcal{L}_2$. The term $\mathcal{R}(L_1, L_2)$ encourages alignment of representations of words that are translations. Training an embedding set $E_J = E_{L_1} \cup E_{L_2}$ is usually done by directly optimizing $\mathcal{L}_J$.

While supervised joint training requires access to parallel resources, recent studies (Lample et al., 2018b; Devlin et al., 2019; Artetxe & Schwenk, 2019; Lample & Conneau, 2019) have suggested that unsupervised joint training without such resources is also effective. Specifically, they show that the cross-lingual regularization term $\mathcal{R}(L_1, L_2)$ does not require direct cross-lingual supervision to achieve highly competitive results. This is because the shared words between $\mathcal{L}_1$ and $\mathcal{L}_2$ can serve implicitly as anchors by sharing their embeddings to ensure that representations of different languages lie in a shared space. Using our notation, the unsupervised joint training approach takes the following steps:
**Step 1:** Construct a joint vocabulary $V_J = V_{L_1} \cup V_{L_2}$ that is *shared* across two languages.
**Step 2:** Concatenate the two training corpora and learn an embedding set $E_J$ corresponding to $V_J$.

The joint vocabulary is composed of three disjoint sets: $V_J^1, V_J^2, V_J^s$, where $V_J^s = V_{L_1} \cap V_{L_2}$ is the shared vocabulary set and $V_J^i$ is the set of tokens that appear in the $i$th language only. Note that a key difference of existing supervised joint training methods is that embeddings corresponding to $V_J^s$ are not shared between $E_{L_1}$ and $E_{L_2}$, meaning that they are disjoint, as in alignment methods.

## 2.3 Discussion

While alignment methods have had great success, there are still some critical downsides, among which we stress the following points:

1. While recent studies in unsupervised joint training have suggested the potential benefits of word sharing, alignment methods rely on two disjoint sets of embeddings. Along with some possible loss of information due to no sharing, one consequence is that finetuning the aligned embeddings on downstream tasks may be sub-optimal due to the lack of cross-lingual constraints at the finetuning stage, whereas shared words can fulfill this role in jointly trained models.

2. A key assumption of alignment methods is the isomorphism of monolingual embedding spaces. However, some recent papers have challenged this assumption, showing that it does not hold for many language pairs (Søgaard et al., 2018; Patra et al., 2019). Also notably, Ormazabal et al. (2019) suggests that this limitation results from the fact that the two sets of monolingual embeddings are independently trained.

On the other hand, the *unsupervised* joint training method is much simpler and doesn't share these disadvantages with the alignment methods, but there are also some key limitations:

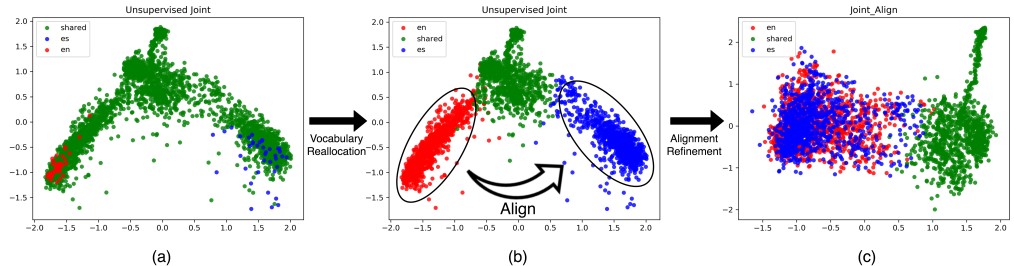

Figure 1: PCA visualization of English and Spanish embeddings learnt by unsupervised joint training as in Lample et al. (2018b). As shown by plots (a) and (b), most words are shared in the initial embedding space but not well-aligned, hence the oversharing problem. Plots (b) and (c) shows that the vocabulary reallocation step effectively mitigates oversharing while the alignment refinement step further improves the poorly aligned embeddings by projecting them into a close neighborhood.

1. It assumes that all shared words across two languages serve implicitly as anchors and thus need not be aligned to other words. Nonetheless, this assumption is not always true, leading to misalignment. For example, the English word "the" will most likely also appear in the training corpus of Spanish, but preferably it should be paired with Spanish words such as "el" and "la" instead of itself. We refer to this problem as oversharing.

2. It does not utilize any explicit form of seed dictionary as in alignment methods, resulting in potentially less accurate alignments, especially for words that are not shared.

Lastly, while the *supervised* joint training approach does not have the same issues of unsupervised joint training, it shares limitation 1 of the alignment methods.

We empirically compare both joint training and alignment approaches in Section 4 and shed light on some of these pros and cons for both paradigms (See Section 4.3.1).

## 3    PROPOSED FRAMEWORK

Motivated by the pros and cons of both paradigms, we propose a unified framework that first uses unsupervised joint training as a coarse initialization and then applies alignment methods for refinement, as demonstrated in Figure 1. Specifically, we first build a single set of embeddings with a shared vocabulary through unsupervised joint training, so as to alleviate the limitations of alignment methods. Next, we use a vocabulary reallocation technique to mitigate oversharing, before finally resorting back to alignment methods to further improve the embeddings' quality. Lastly, we show that this framework can generalize to contextualized representations.

### 3.1    UNIFYING ALIGNMENT WITH JOINT TRAINING

Our proposed framework mainly involves three components and we discuss each of them as follows.

**Joint Initialization.**    We use unsupervised joint training (Lample et al., 2018b) to train the initial CLWE. As described in Section 2.2, we first obtain a joint vocabulary $V_J$ and train its corresponding set of embeddings $E_J$ on the concatenated corpora of two languages. This allows us to obtain a single set of embeddings that maximizes sharing across two languages. To train embeddings, we used fastText[4] (Bojanowski et al., 2017) in all our experiments for both word and subword tokens.

**Vocabulary Reallocation.**    As discussed in Section 2.3, a key issue of unsupervised joint training is oversharing, which prohibits further refinement as shown in Figure 1. To alleviate this drawback, we attempt to "unshare" some of the overshared words, so their embeddings can be better aligned in the next step. Particularly, we perform a vocabulary reallocation step such that words appearing

---

[4]https://github.com/facebookresearch/fastText

mostly exclusively in the $i$th language are reallocated from the shared vocabulary $V_J^s$ to $V_J^i$, whereas words that appear similarly frequent in both languages stay still in $V_J^s$. Formally, for each token $w$ in the shared vocabulary $V_J^s$, we use the ratio of counts within each language to determine whether it belongs to the shared vocabulary:

$$r = \frac{T_{L_2}}{T_{L_1}} \cdot \frac{C_{L_1}(w)}{C_{L_2}(w)}, \tag{5}$$

where $C_{L_i}(w)$ is the count of $w$ in the training corpus of the $i$th language and $T_{L_i} = \sum_w C_{L_i}(w)$ is the total number of tokens. The token $w$ is allocated to the shared vocabulary if

$$\frac{1-\gamma}{\gamma} \leq r \leq \frac{\gamma}{1-\gamma}, \tag{6}$$

where $\gamma$ is a hyper-parameter. Otherwise, we put $w$ into either $V_J^1$ or $V_J^2$, where it appears mostly frequent. The above process generates three new disjoint vocabulary sets $V_J^{1'}, V_J^{2'}, V_J^{s'}$ and their corresponding embeddings $E_J^{1'}, E_J^{2'}, E_J^{s'}$ that are used thereafter. Note that, $V_J' = V_J$ and $E_J' = E_J$.

**Alignment Refinement.** The unsupervised joint training method does not explicitly utilize any dictionary or form of alignment. Thus, the resulting embedding set is coarse and ill-aligned in the shared vector space, as demonstrated in Figure 1. As a final refinement step, we utilize any off-the-shelf alignment method to refine alignments across the non-sharing embedding sets, i.e. mapping $E_J^{1'}$ to $E_J^{2'}$ and leaving $E_J^{s'}$ untouched. This step could be conducted by either supervised or unsupervised alignment method and we compare both in our experiments.

### 3.2 Extension to Contextualized Representations

As our framework is highly generic and applicable to any alignment and unsupervised joint training methods, it can naturally generalize to contextualized word representations by aligning the fixed outputs of a multilingual encoder such as multilingual BERT (M-BERT) (Devlin et al., 2019). While our vocab reallocation technique is no longer necessary as contextualized representations are dependent on context and thus dynamic, we can still apply alignment refinement on extracted contextualized features for further improvement. For instance, as proposed by Aldarmaki & Diab (2019), one method to perform alignment on contextualized representations is to first use word alignment pairs extracted from parallel corpora as a dictionary, learn an alignment matrix $W$ based on it, and apply $W$ back to the extracted representations. To obtain $W$, we can solve Eq.( 1) as described in Section 2.1, where the embedding matrices $X_{L_1}$ and $X_{L_2}$ now contain contextualized representations of aligned word pairs. Note that this method is applicable to fixed representations but not finetuning.

## 4 Experiments

We evaluate the proposed approach and compare with alignment and joint training methods on three NLP benchmarks. This evaluation aims to: (1) systematically compare alignment vs. joint training paradigms and reveal their pros and cons discussed in Section 2.3, (2) show that the proposed framework can effectively alleviate limitations of both alignment and joint training, and (3) demonstrate the effectiveness of the proposed framework in both non-contextualized and contextualized settings.

### 4.1 Evaluation Tasks

**Bilingual Lexicon Induction (BLI)** This task has been the *de facto* evaluation task for CLWE methods. It considers the problem of retrieving the target language translations of source langauge words. We use bilingual dictionaries complied by Conneau et al. (2018a) and test on six diverse language pairs, including Chinese and Russian, which use a different writing script than English. Each test set consists of 1500 queries and we report *precision at 1* scores (P@1), following standard evaluation practices (Conneau et al., 2018a; Glavas et al., 2019).

**Name Entity Recognition (NER)** We also evaluate our proposed framework on cross-lingual NER, a sequence labeling task, where we assign a label to each token in a sequence. We evaluate both

| | en-es | es-en | en-fr | fr-en | en-de | de-en | en-it | it-en | en-ru | ru-en | en-zh | zh-en | avg |
|---|---|---|---|---|---|---|---|---|---|---|---|---|---|
| Alignment Methods | | | | | | | | | | | | | |
| (1) MUSE (Conneau et al., 2018a) | 81.7 | 83.3 | 82.3 | 82.1 | 74.0 | 72.0 | 77.7 | 78.2 | 44.0 | 59.1 | 32.5 | 31.4 | 66.5 |
| (2) VECMAP (Artetxe et al., 2018a) | 82.3 | 84.7 | 82.3 | 83.6 | 75.1 | 74.3 | - | - | 49.2 | 65.6 | 0.0 | 0.0 | - |
| (3) DeMa-BWE (Zhou et al., 2019) | 82.8 | 84.9 | 83.1 | 83.5 | 77.2 | 74.4 | - | - | 49.2 | 65.7 | 42.5 | 37.9 | - |
| (4) Procrustes (Smith et al., 2017) | 81.4 | 82.9 | 81.1 | 82.4 | 73.5 | 72.4 | 77.5 | 77.9 | 51.7 | 63.7 | 42.7 | 36.7 | 68.7 |
| (5) GeoMM (Jawanpuria et al., 2019) | 81.4 | 85.5 | 82.1 | 84.1 | 74.7 | 76.7 | 77.9 | 80.9 | 51.3 | 67.6 | 49.1 | 45.3 | 71.4 |
| (6) RCSLS (Joulin et al., 2018b) | 84.1 | 86.3 | 83.3 | 84.1 | 79.1 | 76.3 | 78.5 | 79.8 | 57.9 | 67.2 | 45.9 | 46.4 | 72.4 |
| (7) RCSLS + IN (Zhang et al., 2019) | 83.9 | - | **83.9** | - | 78.1 | - | 79.1 | - | 57.9 | - | 48.6 | - | - |
| Joint Traing Methods | | | | | | | | | | | | | |
| (8) Unsupervised Joint | 33.4 | 36.6 | 42.2 | 47.4 | 39.5 | 41.4 | 36.8 | 38.8 | 4.0 | 3.5 | 17.9 | 10.2 | 29.3 |
| (9) Supervised Joint (Duong et al., 2016) | 79.7 | 79.8 | 78.1 | 76.7 | 67.5 | 68.9 | 74.4 | 74.1 | 41.8 | 51.8 | 46.7 | 43.3 | 65.2 |
| (10) Joint - Replace | 48.2 | 47.7 | 49.4 | 52.1 | 46.5 | 46.9 | 43.8 | 45.8 | 20.3 | 36.6 | 32.7 | 34.1 | 42.0 |
| Joint Align Framework | | | | | | | | | | | | | |
| (11) Joint_Align (w/o AR) | 55.9 | 62.8 | 61.8 | 67.0 | 49.1 | 54.6 | 50.2 | 51.4 | 8.7 | 8.2 | 19.4 | 18.2 | 42.3 |
| (12) Joint_Align + MUSE | 81.4 | 84.2 | 82.8 | 83.6 | 74.2 | 72.2 | 77.5 | 81.5 | 45.0 | 58.3 | 36.1 | 35.3 | 67.7 |
| (13) Joint_Align + RCSLS (w/o VR) | 34.2 | 37.0 | 41.2 | 46.8 | 34.0 | 35.6 | 35.3 | 35.1 | 7.7 | 5.2 | 20.2 | 15.7 | 29.0 |
| (14) Joint_Align + GeoMM | 82.6 | 85.7 | 82.5 | 84.2 | 75.5 | 77.2 | 78.2 | 81.4 | 52.4 | 67.7 | 50.4 | 46.5 | 72.0 |
| (15) Joint_Align + RCSLS | **84.7** | **87.9** | 83.5 | **85.6** | **79.6** | **78.0** | **80.6** | **84.0** | **59.8** | **67.8** | **54.3** | **48.7** | **74.5** |

Table 1: **Precision@1 for the BLI task on the MUSE dataset**[6]. Within each category, unsupervised methods are listed at the top while supervised methods are at the bottom. The best result for unsupervised methods is underlined while **bold** signifies the overall best. "IN" refers to iterative normalization proposed in Zhang et al. (2019), "AR" refers to alignment refinement and "VR" refers to vocabulary reallocation.

non-contextualized and contextualized word representations on the CoNLL 2002 and 2003 benchmarks (Tjong Kim Sang, 2002; Tjong Kim Sang & De Meulder, 2003), which contain 4 European languages. To measure the quality of CLWE, we perform zero-shot cross-lingual classification, where we train a model on English and directly apply it to each of the other 3 languages.

**Unsupervised Machine Translation (UMT)** Lastly, we test our approach using the unsupervised MT task, on which the initialization of CLWE plays a crucial role (Lample et al., 2018b). Note that our purpose here is to directly compare with similar studies in Lample et al. (2018b), and thus we follow their settings and consider two language pairs, English-French and English-German, and evaluate on the widely used WMT'14 en-fr and WMT'16 en-de benchmarks.

## 4.2 EXPERIMENTAL SETUP

For the BLI task, we compare our framework to recent state-of-the-art methods. We obtain numbers from the corresponding papers or Zhou et al. (2019), and use the official tools for MUSE (Conneau et al., 2018a), GeoMM (Jawanpuria et al., 2019) and RCSLS (Joulin et al., 2018b) to obtain missing results. We consider the method of Duong et al. (2016) for supervised joint training based on bilingual dictionaries, which is comparable to supervised alignment methods in terms of resources used. For unsupervised joint training, we train uncased joint fastText word vectors of dimension 300 on concatenated Wikipedia corpora of each language pair with default parameters. The hyperparameter $\gamma$ is selected from $\{0.7, 0.8, 0.9, 0.95\}$ on validation sets. For the alignment refinement step in our proposed framework, we use **RCSLS** and **GeoMM** to compare with supervised methods, and **MUSE** for unsupervised methods. In addition, we include an additional baseline of joint training, denoted as **Joint - Replace**, which is identical to unsupervised joint training except that it utilizes a seed dictionary to randomly replace words with their translations in the training corpus. Following standard practices, we consider the top 200k most frequent words and use the cross-domain similarity local scaling (CSLS) (Conneau et al., 2018a) as the retrieval criteria. Note that a concurrent work (Artetxe et al., 2019) proposed a new retrieval method based on MT systems and produced state-of-the-art results. Although their method is applicable to our framework, it has high computational cost and is out of the scope of this work.

For the NER task: (1) For non-contextualized representations, we train embeddings the same way as in the BLI task and use a vanilla Bi-LSTM-CRF model (Lample et al., 2016). For all alignment steps, we apply the supervised Procrustes method using dictionaries from the MUSE library for simplicity. (2) For contextualized representations, we use M-BERT, an unsupervised joint training model, as

---

[6]We found that the official evaluation script of MUSE ignores test pairs that are out-of-vocabulary (OOV) on the source or the target sides, which could occur occasionally when evaluating our model due to the proposed vocabulary reallocation step. To ensure fair comparison, we include these OOV pairs. Specifically, if the source word is OOV, we retrieve itself; otherwise, we count the pair as incorrect. Please see appendix B for details.

| | en-es | es-en | en-fr | fr-en | en-de | de-en | en-it | it-en | en-ru | ru-en | en-zh | zh-en | avg |
|---|---|---|---|---|---|---|---|---|---|---|---|---|---|
| Unsupervised | | | | | | | | | | | | | |
| (1) MUSE (Conneau et al., 2018a) | 77.1 | 82.5 | 76.4 | 78.0 | 67.4 | 67.8 | 72.5 | 77.5 | 42.7 | 50.2 | 28.7 | 29.1 | 62.5 |
| (2) Unsupervised Joint | 3.7 | 10.2 | 5.1 | 10.7 | 8.5 | 10.5 | 7.8 | 8.1 | 0.4 | 2.7 | 2.5 | 6.4 | 6.4 |
| (3) Joint_Align + MUSE | 77.5 | 83.0 | 77.0 | 79.5 | 66.7 | 68.0 | 70.9 | 78.0 | 43.5 | 55.1 | 32.3 | 32.7 | 63.7 |
| Supervised | | | | | | | | | | | | | |
| (4) RCSLS (Joulin et al., 2018b) | 78.0 | 83.9 | 76.0 | 78.6 | 68.2 | 68.4 | 71.8 | 78.2 | 50.7 | 56.9 | 51.0 | 41.7 | 67.0 |
| (5) Supervised Joint (Duong et al., 2016) | 76.8 | 80.8 | 73.4 | 76.1 | 60.1 | 61.7 | 69.7 | 76.2 | 41.0 | 51.8 | **52.3** | 43.3 | 63.6 |
| (6) Joint_Align + RCSLS | **82.1** | **84.6** | **78.1** | **80.4** | **68.4** | **70.4** | **73.7** | **79.0** | **59.0** | **66.8** | 51.4 | **45.7** | **70.0** |

Table 2: **Precision@1 for the BLI task on the MUSE dataset with test pairs of same surface form removed**. The best result for unsupervised methods is underlined while **bold** signifies the overall best.

our base model and apply our proposed framework on it by first aligning its extracted features and then feeding them to a task-specific model (M-BERT Feature + Align). Specifically, we use the sum of the last 4 M-BERT layers' outputs as the extracted features. To obtain the alignment matrices, one for each layer, we use 30k parallel sentences from the Europarl corpus for each language pair and follow the procedure of Section 3.2. We feed the extracted features as inputs to a task-specific model with 2 Bi-LSTM layers and a CRF layer (see appendix A). We compare our framework to both finetuning (M-BERT Finetune), which has been studied by previous papers, and feature extraction (M-BERT Feature). Lastly, we also compare against XLM, a supervised joint training model.

For the UMT task, we use the exact same data, architecture and parameters released by Lample et al. (2018b)[7]. We simply use different embeddings trained with the same data as inputs to the model.

## 4.3 Results and Analysis

### 4.3.1 Alignment vs. Joint Training

We compare alignment methods with joint training on all three downstream tasks. As shown in Table 1 and Table 3, we find alignment methods significantly outperform the joint training approach by a large margin in all language pairs for both BLI and NER. However, the unsupervised joint training method is superior than its alignment counterpart on the unsupervised MT task as demonstrated in 2(c). While these results demonstrate that their relative performance is task-dependent, we conduct further analysis to reveal three limitations as discussed in Sec 2.3.

First, their poor performance on BLI and NER tasks shows that unsupervised joint training fails to generate high-quality alignments due to the lack of a fine-grained seed dictionary as discussed in its limitation 2. To evaluate accuracy on words that are not shared, we further remove test pairs of the same surface form (e.g. (hate, hate) as a test pair for en-de) of the BLI task and report their results in Table 2. We find unsupervised joint training (row 2) to achieve extremely low scores which shows that emebddings of non-shared parts are poorly aligned, consistent with the PCA visualization shown in Figure 1.

Moreover, we delve into the relative performance of the two paradigms on the MT task by plotting their test BLEU scores of the first 20 epochs in Figure 2(a) and 2(b). We observe that the alignment method actually obtains *higher* BLEU scores in the first few epochs, but gets surpassed by joint training in later epochs. This shows the importance of parameter sharing as discussed in limitation 1 of alignment methods: shared words can be used as a cross-lingual constraint for unsupervised joint training during fine-tuning but this constraint cannot easily be used in alignment methods. The lack of sharing is also a limitation for the supervised joint training method, which performs poorly on the MT task even with supervision as shown in Figure 2(c).

Lastly, we demonstrate that oversharing can be sub-optimal for unsupervised joint training as discussed in its limitation 2. Specifically, we conduct ablation studies for our framework in Table 1. Applying alignment refinement on unsupervised joint training without any vocabulary reallocation does not improve its performance (row 13). On the other hand, simple vocabulary reallocation alone boosts the performance by quite a margin (row 11). This shows some words are shared erroneously across languages in unsupervised joint training, thereby hindering its performance. In addition, while utilizing a seed dictionary (row 10) improves the performance of unsupervised joint train-

---

[7]https://github.com/facebookresearch/UnsupervisedMT

| | es | nl | de | avg |
|---|---|---|---|---|
| **Non-contextualized** | | | | |
| Unsupervised Joint | 50.28 | 42.77 | 21.49 | 38.18 |
| Supervised Joint (Duong et al., 2016) | 63.16 | 63.60 | 36.24 | 54.33 |
| Joint - Replace | 65.28 | 68.44 | 51.59 | 61.77 |
| Align | 69.00 | 71.33 | 52.17 | 64.17 |
| Joint_Align | 70.46 | 72.10 | 56.47 | 66.34 |
| Xie et al. (2018)[‡] | 71.67 | 70.90 | 57.43 | 66.67 |
| Chen et al. (2019)[‡] | 73.50 | 72.40 | 56.00 | 67.30 |
| **Contextualized** | | | | |
| XLM Finetune (Lample & Conneau, 2019)[*] | 63.18 | - | 67.55 | - |
| M-BERT Finetune (Pires et al., 2019) | 73.59 | 77.36 | 69.74 | 73.56 |
| M-BERT Finetune (Wu & Dredze, 2019) | 74.96 | 77.57 | 69.56 | 74.03 |
| M-BERT Finetune (Keung et al., 2019) | 75.00 | 77.50 | 68.60 | 73.70 |
| M-BERT Finetune + Adv (Keung et al., 2019) | 74.30 | 77.60 | **71.90** | 74.60 |
| M-BERT Feature | 74.23 | 78.65 | 67.63 | 73.50 |
| M-BERT Feature + Align | **75.77** | **79.03** | 70.54 | **75.11** |

Table 3: **F1 score for the cross-lingual NER task.** "Adv" refers to adversarial training. [‡] denotes results that are not directly comparable due to different resources and architectures used. [*] denotes supervised XLM model trained with MLM and TLM objectives. Its Dutch (nl) result is blank because the model is not pretrained on it. **Bold** signifies state-of-the-art results. We report the average of 5 runs.

ing, it still suffers from the oversharing problem and performs worse compared to supervised joint training (row 9).

### 4.3.2 Evaluation of Proposed Framework

As shown in Table 1, Table 3, and Figure 2, our proposed framework substantially improves over the alignment and joint training baselines on all three tasks. In particular, it outperforms existing methods on all language pairs for the BLI task (using the CSLS as retrieval metric) and achieves state-of-the-art results on 2 out of 3 language pairs for the NER task. Besides, we show that it alleviates limitations of alignment and joint training methods shown in the previous section.

First, the proposed framework largely improves the poor alignment of unsupervised joint training, especially for non-sharing parts. As shown in Table 1, the proposed Joint_Align framework achieves comparable results to prior methods in the unsupervised case (row 12) and it outperforms previous state-of-the-art methods in the supervised setting (row 15). Specifically, our proposed framework can generate well-aligned embeddings after alignment refinement is applied to the initially ill-aligned embeddings, as demonstrated in Figure 1. This is further verified by results in Table 2, where our proposed framework largely improves accuracy on words not shared between two languages over the unsupervised joint training baseline (row 3 and 6 vs row 2).

Besides, our ablation study in Table 1 further shows the effectiveness of the proposed vocabulary reallocation technique, which alleviates the issue of oversharing. Particularly, we observe no improvement compared to unsupervised joint training baseline (row 8) when an alignment refinement step is used without vocabulary reallocation (row 13), while a vocabulary reallocation step alone significantly improves the performance (row 11). This is consistent with Figure 1 and shows that the oversharing is a bottleneck for applying alignment methods to joint training. It also suggests detecting what to share is crucial to achieve better cross-lingual transfer.

Lastly, while supervised joint training shares the limitation 1 of alignment methods and performs poorly when finetuned, our proposed framework exploits the same idea of vocabulary sharing used in unsupervised joint training. In the MT tasks, our framework obtains a maximum gain of 2.97 BLEU over baselines we ran and consistently performs better than results reported in Lample et al. (2018b). In addition, Figure 2 shows that Joint_Align not only converges faster in earlier training epochs but also consistently outperforms the two baselines thereafter. These empirical findings demonstrate the effectiveness of our proposed methods in the non-contextualized case.

### 4.3.3 Contextualized Word Representations

As can be seen in Table 3, when using our framework (M-BERT Feature + Align), we achieve state-of-the-art results on cross-lingual NER on 2 out of 3 languages and the overall average. This shows that our framework can effectively generalize to contextualized representations. Specifically, our framework improves over both the M-BERT feature extraction and finetuning baselines on all three language pairs. However, when compared to non-contextualized results, the gain of using alignment refinement on top of unsupervised joint training is much smaller. This suggests that, as the contextualized unsupervised joint training model performs very well already even without

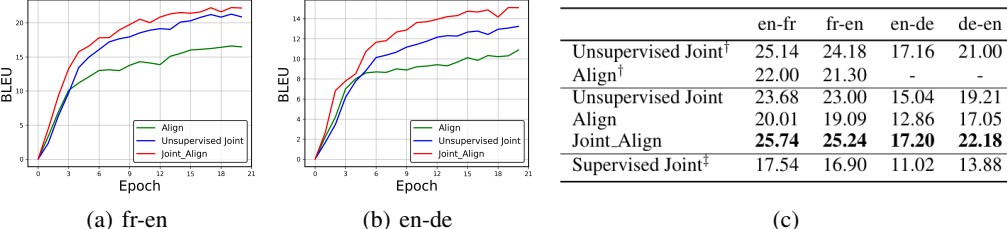

|  | en-fr | fr-en | en-de | de-en |
|---|---|---|---|---|
| Unsupervised Joint[†] | 25.14 | 24.18 | 17.16 | 21.00 |
| Align[†] | 22.00 | 21.30 | - | - |
| Unsupervised Joint | 23.68 | 23.00 | 15.04 | 19.21 |
| Align | 20.01 | 19.09 | 12.86 | 17.05 |
| Joint_Align | **25.74** | **25.24** | **17.20** | **22.18** |
| Supervised Joint[‡] | 17.54 | 16.90 | 11.02 | 13.88 |

(a) fr-en  (b) en-de  (c)

Figure 2: (a)(b): **Results on MT of Align, Joint and our framework for the first 20 training epochs.** Results after 20 epochs have similar patterns. (c): **BLEU scores for the MT task.** Results evaluated on the WMT'14 English-French and WMT'16 German-English. All training settings are the same for each language pair except the embedding initialization. Note that we are not trying to outperform state-of-the-art methods (Song et al., 2019) but rather to observe improvements afforded by embedding initialization. [†]Results reported by Lample et al. (2018b). Our results are obtained using the official code released by the author. [‡]Duong et al. (2016) is a supervised method that we include for analysis purpose only and is not directly comparable to other results in this table.

any supervision, it is harder to achieve large improvements. While alignment refinement relies on word alignment, a process that is noisy itself, a better alignment approach may be warranted. Lastly, the reason why a supervised joint training model, XLM, performs worse than its unsupervised counterpart, M-BERT, is likely that XLM uses an uncased vocabulary, where casing information is important for NER tasks.

## 5 RELATED WORK

Word embeddings (Mikolov et al., 2013a; Ruder et al., 2019) are a key ingredient to achieving success in monolingual NLP tasks. However, directly using word embeddings independently trained for each language may cause negative transfer (Wang et al., 2019) in cross-lingual transfer tasks. In order to capture the cross-lingual mapping, a rich body of existing work relying on cross-lingual supervision, including bilingual dictionaries (Mikolov et al., 2013a; Faruqui & Dyer, 2014; Artetxe et al., 2016; Xing et al., 2015; Duong et al., 2016; Gouws & Søgaard, 2015; Joulin et al., 2018a), sentence-aligned corpora (Kočiský et al., 2014; Hermann & Blunsom, 2014; Gouws et al., 2015) and document-aligned corpora (Vulić & Moens, 2016; Søgaard et al., 2015).

Besides, unsupervised alignment methods aim to eliminate the requirement for cross-lingual supervision. Early work of Cao et al. (2016) matches the mean and the standard deviation of two embedding spaces after alignment. Barone (2016); Zhang et al. (2017a;b); Conneau et al. (2018a) adapted a generative adversarial network (GAN) (Goodfellow et al., 2014) to make the distributions of two word embedding spaces indistinguishable. Follow-up works improve upon GAN-based training for better stability and robustness by introducing Sinkhorn distance (Xu et al., 2018), by stochastic self-training (Artetxe et al., 2018a), or by introducing latent variables (Dou et al., 2018).

While alignment methods utilize embeddings trained independently on different languages, joint training methods train word embeddings at the same time. Klementiev et al. (2012) train a bilingual dictionary-based regularization term jointly with monolingual language model objectives while Kočiský et al. (2014) defines the cross-lingual regularization with the parallel corpus. Another branch of methods (Xiao & Guo, 2014; Gouws & Søgaard, 2015; Ammar et al., 2016; Duong et al., 2016) build a pseudo-bilingual corpus by randomly replacing words in monolingual corpus with their translations and use monolingual word embedding algorithms to induce bilingual representations. The unsupervised joint method by Lample & Conneau (2019) simply exploit words that share the same surface form as bilingual "supervision" and directly train a shared set of embedding with joint vocabulary. Recently, unsupervised joint training of contextualized word embeddings through the form of multilingual language model pretraining using shared subword vocabularies has produced state-of-the-art results on various benchmarks (Devlin et al., 2019; Artetxe & Schwenk, 2019; Lample & Conneau, 2019; Pires et al., 2019; Wu & Dredze, 2019).

A concurrent work by Ormazabal et al. (2019) also compares alignment and joint method in the bilingual lexicon induction task. Different from their setup which only tests on supervised settings, we conduct analysis across various tasks and experiment with both supervised and unsupervised conditions. While Ormazabal et al. (2019) suggests that the combination of the alignment and joint model could potentially advance the state-of-art of both worlds, we propose such a framework and empirically verify its effectiveness on various tasks and settings.

## 6 CONCLUSION

In this paper, we systematically compare the alignment and joint training methods for CLWE. We point out that the nature of each category of methods leads to certain strengths and limitations. The empirical experiments on extensive benchmark datasets and various NLP tasks verified our analysis. To further improve the state-of-art of CLWE, we propose a simple hybrid framework which combines the strength from both worlds and achieves significantly better performance in the BLI, MT and NER tasks. Our work opens a promising new direction that combines two previously exclusive lines of research. For future work, an interesting direction is to find a more effective word sharing strategy.

**Acknowledgments:** This research was sponsored by Defense Advanced Research Projects Agency Information Innovation Office (I2O) under the Low Resource Languages for Emergent Incidents (LORELEI) program, issued by DARPA/I2O under Contract No. HR0011-15-C0114. The views and conclusions contained in this document are those of the authors and should not be interpreted as representing the official policies, either expressed or implied, of the U.S. government. The U.S. government is authorized to reproduce and distribute reprints for government purposes notwithstanding any copyright notation here on.

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

APPENDIX

## A  NER EXPERIMENT DETAILS

Here we include some additional details for the NER experiments with contextualized representations:

- **Alignment** As described in Section 3.2, we apply word alignment methods, such as fastalign (Dyer et al., 2013), on parallel data to extract word-aligned pairs for learning the alignment matrix. As M-BERT is based on subword tokens, we use the average of the representations of all subword tokens that correspond to a word as the representation for that word. For instance, assume that an English word "Resumption" is aligned to a German word "Wiederaufnahme", and they are tokenized by M-BERT as "Res", "##sumption", and "Wie", "##dera", "##uf", "##nahme", respectively. Then the representation for "Resumption" is the average of the representations of subword tokens "Res" and "##sumption", and the same goes for "Wiederaufnahme".

- **Hyperparameters** For the task-specific NER model, we use a 2-layer Bi-LSTM with a hidden size of 768 followed by a CRF layer. We apply a dropout rate of 0.5 on the input

| | en-es | es-en | en-fr | fr-en | en-de | de-en | en-it | it-en | en-ru | ru-en | en-zh | zh-en | avg |
|---|---|---|---|---|---|---|---|---|---|---|---|---|---|
| | | | | | Alignment Methods | | | | | | | | |
| (1) MUSE (Conneau et al., 2018a) | 82.1 | 83.5 | 82.6 | 83.1 | 74.1 | 71.8 | 77.4 | 79.8 | 44.1 | 59.1 | 34.1 | 31.6 | 66.9 |
| (4) Procrustes (Smith et al., 2017) | 81.6 | 82.0 | 80.5 | 81.5 | 74.2 | 73.3 | 77.0 | 77.8 | 50.8 | 63.5 | 44.2 | 37.0 | 68.6 |
| (5) GeoMM (Jawanpuria et al., 2019) | 82.1 | 86.9 | 82.3 | 85.1 | 75.3 | 77.2 | 78.6 | 82.2 | 51.7 | 67.8 | 50.5 | 45.6 | 72.1 |
| (6) RCSLS (Joulin et al., 2018b) | 82.8 | 84.3 | 82.4 | 83.3 | 78.6 | 75.6 | 78.3 | 81.0 | 57.7 | 66.8 | 48.3 | 45.6 | 72.1 |
| | | | | | Joint Traing Methods | | | | | | | | |
| (7) Unsupervised Joint | 33.2 | 36.3 | 41.5 | 46.8 | 39.1 | 40.7 | 35.8 | 38.1 | 4.1 | 3.7 | 8.2 | 5.7 | 27.8 |
| (8) Supervised Joint (Duong et al., 2016) | 80.1 | 80.5 | 78.6 | 77.1 | 67.2 | 68.3 | 74.6 | 74.5 | 41.7 | 51.8 | 47.2 | 44.0 | 65.5 |
| | | | | | Joint Align Framework | | | | | | | | |
| (9) Joint_Align (w/o AR) | 56.8 | 63.2 | 62.2 | 67.2 | 49.2 | 55.1 | 50.6 | 51.9 | 8.7 | 8.2 | 19.5 | 18.4 | 42.6 |
| (10) Joint_Align + MUSE | 82.4 | 85.0 | 83.5 | 84.7 | 74.6 | 72.9 | 78.2 | 82.6 | 46.1 | 58.7 | 39.9 | 36.2 | 68.7 |
| (11) Joint_Align + RCSLS (w/o VR) | 34.3 | 36.8 | 41.0 | 47.0 | 34.3 | 35.9 | 35.5 | 35.2 | 7.6 | 5.2 | 21.3 | 16.1 | 29.2 |
| (12) Joint_Align + GeoMM | 83.9 | 86.2 | 83.1 | 85.2 | 76.1 | 77.9 | 79.0 | 82.8 | 53.7 | 68.3 | 54.8 | 48.0 | 73.3 |
| (13) Joint_Align + RCSLS | **87.1** | **88.5** | **84.2** | **86.6** | **80.1** | **78.7** | **81.3** | **85.2** | **61.3** | **68.3** | **59.6** | **50.7** | **76.0** |

Table 4: **Precision@1 for the BLI task on the MUSE dataset using test set produced by vocabulary reallocation**. Within each category, unsupervised methods are listed at the top while supervised methods are at the bottom. **Bold** signifies the overall best results. "AR" refers to alignment refinement and "VR" refers to vocabulary reallocation.

and the output of the Bi-LSTM, and use Adam with default parameters and a learning rate of 0.0001 for optimization. We train the model for 40 epochs with a batch size of 10, and evaluate the model per 150 steps. For prediction, we feed the outputs of the Bi-LSTM that correspond to the first subword tokens of each word to the CRF model. This is identical to finetuning BERT on the NER task, except that in our case the outputs that correspond to the first subword token are fed into a CRF, rather than a linear layer as done in BERT.

# B BLI TEST PAIRS

The official evaluation script[8] of MUSE only includes test pairs whose source words and target words both appear in their corresponding vocabularies, leaving out those that are OOV on either side. Since our proposed vocabulary reallocation step modifies both the source and target vocabularies, the script may exclude some test pairs when evaluating our model. For example, the word "age" from the test pair (age, age) for *en-fr* could be allocated as an *en* (not *shared*) word, so it is OOV on the *fr* side and the pair would thus be left out by the script. As a result, the total number of test pairs would be smaller, thereby changing the denominator when we calculate accuracy. To ensure fair comparison, we include these OOV pairs so the total number of test pairs stays the same. Specifically, if the source word of a test pair is OOV, we retrieve itself. Otherwise, we count it as incorrect.

In addition, we further investigate which pairs are left out and found that the MUSE benchmark contains some noisy test data. Specifically, we find that the majority of these pairs are in the same surface form, such as (sit, sit), but many target words are not actual translations of the source words. For example, we found test pairs such as {(age, age), (century, century)} for *en-fr* and {(mickey, mickey), (uncredited, uncredited)} for *en-zh*. Clearly, these are English words and should not be considered as appropriate translations for French or Chinese. In Table 1, we mark these pairs as incorrect for our framework to ensure fair comparison, while in fact our framework *correctly* allocates these words to English. To reveal the full picture, we also conduct BLI experiments without test pairs that got left out due to vocabulary reallocation. The results are shown in Table 4. We observe that our proposed framework obtains a gain of 3.9 accuracy on average over the RCSLS baseline.

---

[8]https://github.com/facebookresearch/MUSE/blob/master/src/evaluation/word_translation.py.

