# OpenReview forum: "Cross-lingual Alignment vs Joint Training: A Comparative Study and A Simple Unified Framework"
_ICLR.cc/2020/Conference — Accept (Poster)_

### Official Review · AnonReviewer3 · 2019-10-21
**Official Blind Review #3**

**Rating:** 8

**Review:**

This paper compares to approaches to bilingual lexicon induction, one which relies on joint training and the other which relies on projecting two languages' representations into a shared space. It also shows which method performs better on which of three tasks (lexicon induction, NER and MT). The paper includes an impressive number of comparisons and tasks which illuminate these two popular methods, making this a useful contribution. It also compares contextualized to non-contextualized embeddings.

I have some questions:
-can you goldilocks the amount of sharing in an alignment method? Put a different way, how much is performance affected if you perform alignment w/variously sized sub-partitions of the seed dictionary? Can you find a core lexicon (perhaps most common words shared cross-lingually) that will provide a good-enough alignment to joint-align with? If you were very ambitious, you could try artificially vary the amount of lexicon disjointness across languages (for camera-ready) and report how much your results are affected by incomplete overlap in translation variants.
-for Table 1, do you have any ideas about why certain languages do better on eng--> them and others do better on them-->eng?
-do you have any analysis exploring what is shared? wrt this sentence "It also suggests detecting what to share is crucial to achieve better cross-lingual transfer."

Please address:
- I would like more intuitive motivation for (6).

Small notes:
-Fig. 1 font is too small, you could also remove excess axes. There's also overlap between the labeled arrow and the y-axis label btw (a) and (b).
-"oversharing" should be typographically delimited as a definition in "We refer to this problem as oversharing"
-typos: bottom of p4: "Secition 2.3"
-"e.g." or "i.e." should be followed by a ","
-list which languages you use on bottom of p5
-Table 3 caption looks crazy, what happened to your spacing there?

**Experience Assessment:**

I do not know much about this area.

**Review Assessment: Checking Correctness Of Derivations And Theory:**

I did not assess the derivations or theory.

**Review Assessment: Checking Correctness Of Experiments:**

I assessed the sensibility of the experiments.

**Review Assessment: Thoroughness In Paper Reading:**

I made a quick assessment of this paper.

---

> ### Author Response · Authors · 2019-11-15
> **Re: Official Blind Review #3**
>
> Thank you for your comprehensive review and valuable feedback. We address your comments one by one as following:
>
> [Manipulate sharing/analyze what is shared]
>
> We agree that controlling and analyzing the amount of sharing is an interesting direction. However, this is a non-trivial task as we explained in our response to reviewer 2. Manipulating the amount of sharing requires using disjoint lookup tables for embeddings, which is perhaps easier to achieve for contextualized representations using deep models than non-contextualized ones using fasttext. Nonetheless, our experiments in the response to reviewer 2 shows that better seed dictionaries for training fasttext are clearly beneficial. In general, what should be shared and how to share them are still open questions that we plan to explore in the future.
>
> [BLI results]
>
> The two directions of en->others and others->en use different test sets and thus their results are not directly comparable. This is perhaps why their relative performance varies.
>
> [Intuition for (6)]
>
> The intuition here is to reallocate words that predominantly appear in a single language to that language. Lambda is a threshold such that if the ratio falls outside of the range it means the word appears mostly in one language (depending on which side it falls) only.
>
> [Typo and better presentation]
>
> We thank the reviewer for pointing out typos and giving kind suggestions. We will be sure to make changes accordingly.

---

### Official Review · AnonReviewer1 · 2019-10-22
**Official Blind Review #1**

**Rating:** 8

**Review:**

This paper compares two approaches to learn cross-lingual word embeddings: cross-lingual alignment (where separately trained embeddings in different languages are mapped into a shared space) and joint training (which combines the training data in all languages and jointly learns a cross-lingual space). The authors argue that each approach has its own advantages, and propose a "unified framework" that essentially applies them sequentially (first train jointly, and then further align them after the necessary vocabulary reallocation).

I have generally positive feelings about the paper. To be honest, I do not like the way the authors frame their work (e.g. the way the method is motivated in Section 2.3 or calling it a "unified framework"), but the actual method they propose does make sense, the experimental evaluation is solid, and the results are generally convincing.


Strengths:

- Good coverage of related work, including recent publications.

- Thorough evaluation in 3 different tasks, much better than what is common in the area (usually limited to BLI).

- The authors experiment with contextual embeddings in addition to conventional word embeddings.

- Generally convincing results with relevant ablations and reasonable baselines.


Weaknesses:

- I feel that framing this as a "unified framework" for cross-lingual alignment and joint training is going a bit too far. The proposed method is as simple as first doing the joint training and then the cross-lingual alignment (with a special treatment for vocabulary reallocation). This is just a sequential application of two class of methods, and not what I would understand as a "unified framework" (which suggests some form of generalization or at least a closer interaction to me). At the same time, the only "joint training" that the authors explore is training regular embeddings over concatenated monolingual corpora, which as far as I know has only been explored by Lample et al. (2018), and I would not consider as a representative example of this family of methods.

- It is not clear how the different vocabulary spaces are handled in BLI. My understanding is that if the query is in the source vocabulary subset the retrieval is done over the target subset, and if it is in the shared subset it is the same query word that is returned, but this is not clear at all.

- I think that the issue of "oversharing" is magnified in the paper. I understand that this is directly connected to the reallocation step in the proposed method, and it of course makes sense to also map words that predominantly appear in a single language. However, in relation to the downstream tasks themselves, oversharing only seems relevant for BLI, where one needs to delimit the retrieval space and avoid returning the query word (which is of course its own nearest neighbor) unless it actually exists in the target language. This is connected to my previous point, and I think should be discussed in isolation.


Other minor things to improve:

- Please make Figure 2c an actual table instead of an image.

**Experience Assessment:**

I have published in this field for several years.

**Review Assessment: Checking Correctness Of Derivations And Theory:**

N/A

**Review Assessment: Checking Correctness Of Experiments:**

I assessed the sensibility of the experiments.

**Review Assessment: Thoroughness In Paper Reading:**

I read the paper at least twice and used my best judgement in assessing the paper.

---

> ### Author Response · Authors · 2019-11-15
> **Re: Official Blind Review #1**
>
> Thank you for your comprehensive review and valuable feedback. We address your comments one by one as following:
>
> [Terminology and joint training]
>
> While we meant to say this is a method that unifies two previously exclusive approaches by calling it a framework, we agree that it can be stated in a better way. We will rephrase our terminology in an updated version.
>
> The joint training paradigm attracts the community’s growing attention recently, especially for the contextualized representations such as multilingual BERT. There are still many open questions such as why joint training models work and how to further improve them. In addition to the non-contextualized joint training explored by [1], we also conduct experiments for the contextualized representations and show that alignment can still improve pretrained models. This is consistent with (very) recent results [2], which also suggest combining ideas from alignment as a future research direction.
>
> [BLI evaluation]
>
> Your understanding of the evaluation protocol is correct. We concatenate each language’s own subset with the shared subset and use the two concatenated parts for evaluation. This is equivalent to your description since if the query is in the shared part it will automatically retrieve itself by the nature of CSLS. We will make this clear in an updated version.
>
> [Oversharing]
>
> The oversharing issue has three undesirable effects: (1) it results in a poor seed dictionary for the joint initialization (2) it prohibits the application of alignment refinement, and because of which, (3) it hinders performance on downstream tasks by making false assumptions on word translations. While we agree that it is most relevant for BLI in our experiments, recent work [3] also shows less sharing (thus a larger vocab size & more parameters) can be beneficial for contextualized representations on harder downstream tasks such as language understanding and question answering, and we hypothesize this is partly due to oversharing of tokens with different meanings in different languages.
>
> [1] Phrase-Based & Neural Unsupervised Machine Translation. Guillaume Lample, Myle Ott, Alexis Conneau, Ludovic Denoyer, Marc'Aurelio Ranzato. EMNLP 2018
> [2] Emerging Cross-lingual Structure in Pretrained Language Models. Shijie Wu, Alexis Conneau, Haoran Li, Luke Zettlemoyer, Veselin Stoyanov. Preprint 2019 https://arxiv.org/abs/1911.01464
> [3] On the Cross-lingual Transferability of Monolingual Representations. Mikel Artetxe, Sebastian Ruder, Dani Yogatama. Preprint 2019 https://arxiv.org/abs/1910.11856

---

### Official Review · AnonReviewer2 · 2019-10-23
**Official Blind Review #2**

**Rating:** 6

**Review:**

This paper proposes a new and simple framework for learning cross-lingual embeddings that, based on well-supported insights, unifies two standard frameworks: joint learning and alignment-based approaches. While I like and acknowledge the fact that the paper examines these frameworks critically and has also some didactic value, I still have some concerns regarding the current experiments, and would like to see some additional analyses in the paper. Honestly, this type of work would work better as a short ACL/EMNLP paper, as the core methodological contribution is a very simple combination of the existing techniques.

With joint methods, it is true that shared words can be used as implicit bilingual learning signal which gets propagated further in the model. Even in the case of alignment-based methods, it was shown that this signal can assist in learning shared cross-lingual spaces, but: 1) such spaces are of lower-quality than the spaces learned by relying on seed dictionaries (see e.g., the work of Vulic and Korhonen (ACL 2016)), 2) this is useful only for similar language pairs which use the same script. The paper fails to provide an insightful discussion on how the scores differ when we move towards more distant languages. For instance, English-Chinese is included as a more distant language pair, and the combined method seems to work fine, but the reader is left wondering why that happens. The same is true for English-Russian. The paper should definitely provide more information and insights here.

One experiment that would be quite interesting imho is to take seed dictionaries to initialise the joint training method. It will not be unsupervised any more, but it would be very useful to report the differences in results between fully unsupervised joint initialisation (which, as mentioned should work only with more similar languages) and such weakly supervised init: e.g., the method could take all one-to-one translation pairs from the seed dictionary as 'shared words'. It would also be interesting to combine such 'shared words' and words that are really shared (identically spelled words) as initialisation and measure how it affects the results, also in light of differences in language distance. Adding one such experiment would make the paper more interesting.

Some recent strong baselines are missing: e.g., I wonder how the model that does self-learning on top of seed dictionaries (similar to the work of Vulic et al., EMNLP 2019) compares to the proposed method. Also, can we use the same self-learning technique with the combined method here? Would that lead to improved results? Another work I would like to see comparisons to is the work of Zhang et al. (ACL 2019) and despite the fact that the authors explicitly stated that they decided not to compare to the work of Artetxe et al. (ACL 2019) as they see it as concurrent work, I would still like to see that comparison as the model of Artetxe et al. seems very relevant to the presented work.

I do not see the extension to contextualized representations (described in Section 3.2) as a real contribution: this is a very straightforward method to apply an alignment-based method on multilingual BERT representations, and very similar techniques have been probed in previous work (e.g., by Aldarmaki & Diab), only the bilingual signal/dictionary came from parallel data instead.

Finally, as mentioned before, one of the must-have experiments is including more (distant and diverse) language pairs and analysing the results across such language pairs, with an aim to further understand how the levels of sharing, over-sharing, and non-isomorphism affect performance. Also, while the authors state that reduced isomorphism is the key problem of alignment-based methods, I still fail to see how exactly the alignment refinement step does not suffer from that problem? More discussion is needed here.

Other comments:
- It would be useful to also point to the survey paper of Ruder et al. (JAIR 2019) where the difference between alignment-based and joint models is described in more detail and could inform an interested reader beyond the confines of the related work section in this paper. Similarly, differences between joint models and alignment-based models have also been analysed by Upadhyay et al. (ACL 2016); Vulic and Korhonen (ACL 2016).

- I like how Figure 1 clearly visualizes the main intuitions behind the proposed framework, and how mitigating the oversharing problem leads to better alignments. This was very nice.

- In light of the problems with silver standard MUSE datasets (see also the recent work of Kementchedjhieva, EMNLP 2019), I would suggest to run BLI experiments on more language pairs: e.g., there are BLI datasets of Glavas et al available for 28 language pairs.

**Experience Assessment:**

I have published in this field for several years.

**Review Assessment: Checking Correctness Of Derivations And Theory:**

I carefully checked the derivations and theory.

**Review Assessment: Checking Correctness Of Experiments:**

I carefully checked the experiments.

**Review Assessment: Thoroughness In Paper Reading:**

I read the paper thoroughly.

---

> ### Author Response · Authors · 2019-11-15
> **Re: Official Blind Review #2**
>
> Thank you for your comprehensive review and valuable feedback. We address your comments one by one as following:
>
> [Improvements of distant language pairs/Reduced isomorphism]
>
> The source of improvement for jointly training two distant languages is still an open question. As recently shown in [1], joint training is still beneficial even for languages using disjoint vocabulary sets. Although their findings are based on contextualized representations, we observe similar improvements for non-contextualized representations in our experiments and we hypothesize that the improvement is due to the overlap in training corpus resulting in more similar graph structures of embeddings (i.e. suffer less from the reduced isomorphism issue).
>
> To analyze the effect of reduced isomorphism, we compute the eigenvector similarity proposed in [2]. This metric measures the degree of similarity between two embedding graphs and therefore can be used to estimate the degree of isomorphism. In particular, we compute this metric using four embeddings: monolingual fasttext, fasttext aligned with RCSLS, joint training (unsupervised), and joint_align with RCSLS. The results are shown below (smaller values indicates graphs are more similar). In particular, we observe that jointly trained embeddings share more similar graph structures than independently trained counterparts and thus they suffer less from the reduced isomorphism problem. These results also suggest that the alignment refinement step suffers less from this problem due to a better initialization from joint training.
>
>                                   en-es |en-fr|en-de|en-it|en-ru|en-zh| avg
> fasttext                       3.25 | 3.73 | 4.48 | 4.15 | 4.92 | 5.02 | 4.26
> fasttext_RCSLS          2.56 | 2.59 | 2.02 | 2.72 | 3.60 | 2.82 | 2.72
> Unsupervised Joint   2.59 | 1.98 | 2.38 | 2.59 | 2.94 | 2.79 | 2.55
> Joint_Align                  1.48 | 2.00 | 1.75 | 2.45 | 3.07 | 2.68 | 2.24
>
> [Joint training with seed dictionary]
>
> We agree that adding a joint training baseline using seed dictionaries is didactic. However, this is a non-trivial task since joint training works best with fasttext in order to utilize subword information, which does not take an explicit form of seed dictionary nor allow tokens with different surface forms to share same embeddings. Therefore, instead of fundamentally modifying fasttext, we use a simple approach: (1) for each word in the dictionary, we randomly replace its occurrences in the concatenated corpus with its translation 50% of the time and keep it unchanged for the rest, (2) we then train the joint training embeddings as in the paper, (3) for words in the seed dictionary, we take the average of their embeddings and treat them as one single word shared between the two languages. We use the same seed dictionary from MUSE as in other experiments. As pointed out by reviewer 3, manipulating the degree of sharing is an ambitious goal that we plan to study in the future and may include in a later version.
>
> As shown in the table below, we observe that using the seed dictionary as “shared words” improves the performance of unsupervised joint training. However, since this method still treats identically spelled words as shared words (which is implicitly forced by the nature of fasttext), it still suffer from the oversharing problem as we observe further improvements by applying an extra vocabulary reallocation step.
>
> BLI:
>                                  en-es|es-en|en-fr|fr-en |en-de|de-en|en-it|it-en|en-ru|ru-en|en-zh|zh-en|avg
> Unsupervised Joint  33.4| 36.6 | 42.2 | 47.4 | 39.5 | 41.4  | 36.8 | 38.8| 4.00 | 3.50 | 17.9  | 10.2  |29.3
> Replace                      48.2| 47.7 | 49.4 | 52.1 | 46.5 | 46.9  | 43.8 | 45.8| 20.3 | 36.6 | 32.7  | 34.1  |42.0
> Replace+VR               67.1| 68.7 | 66.4 | 68.4 | 62.3 | 64.2  | 57.3 | 60.6| 42.6 | 50.1 | 46.5  | 43.6  |58.2
> Joint_Align                 86.0| 88.5 | 83.9 | 85.8 | 79.3 | 78.7  | 79.9 | 83.1| 60.4 | 69.2 | 57.2  | 50.4  |75.2
>
> NER:
>                                        es     |      nl     |     de     |    avg
> Unsupervised Joint  50.28   |   42.77  |  21.49  |   38.18
> Replace                      65.28   |   68.44  |  51.59  |   61.77
> Joint_Align                 70.46   |   72.10  |  56.47  |   66.34
>
> (The method described above is denoted as “Replace” and we also include baselines from the paper for your convenience. “VR” denotes vocabulary reallocation.)
>
> [Extension to contextualized representations]
>
> We agree that applying alignment methods to contextualized representations is straightforward. The point is to show that further alignment on an already well-trained multilingual model can still improve performance, which has also been shown by [3] and is consistent with findings reported in [1].

---

> > ### Author Response · Authors · 2019-11-15
> > **Re: Official Blind Review #2**
> >
> > Continued (due to space limits).
> >
> > [Missing baselines/citations]
> >
> > Thank you for kindly pointing them out. We will add these baselines and citations in an updated version. However, we also want to point out that our framework is generic and thus we simply cannot test all possible combinations of different methods. For that reason, we choose to leave some for future work.
> >
> > [1] On the Cross-lingual Transferability of Monolingual Representations. Mikel Artetxe, Sebastian Ruder, Dani Yogatama. Preprint 2019 https://arxiv.org/abs/1910.11856
> > [2] On the Limitations of Unsupervised Bilingual Dictionary Induction. Anders Søgaard, Sebastian Ruder, Ivan Vulić. ACL 2018
> > [3] Cross-Lingual BERT Transformation for Zero-Shot Dependency Parsing. Yuxuan Wang, Wanxiang Che, Jiang Guo, Yijia Liu, Ting Liu. EMNLP 2019

---

### Public Comment · ~Mozhi_Zhang1 · 2019-10-03
**Stronger supervised BLI baseline**

Interesting work on cross-lingual word embeddings!

I want to point out that there is a stronger recent baseline for supervised BLI: https://www.aclweb.org/anthology/P19-1307.pdf (by normalizing monolingual embeddings before aligning with RCSLS)

---

> ### Author Response · Authors · 2019-10-07
> **Thank you for your helpful comment!**
>
> Thank you for your interest in our work and helpful feedback. [1] is an interesting work and we will include it in our discussion regarding isomorphism assumption in a later version.
>
> Our framework is highly generic and we also were considering the compatibility of this work with our method. But in doing so we became a bit confused regarding the results reported in [1]. It seems that the reported results for RCSLS in [1], with or without the proposed method applied, are lower than the results in the original RCSLS paper [2]. Do you know why this may be? Note that we have reported the original results from the RCSLS paper, which we were also able to reproduce in our own experiments.
>
> Additionally, do you know if code from [1] is publicly available? This would allow us to deepen our understanding of the details of the method.
>
> [1] Are Girls Neko or Shojo? Cross-Lingual Alignment of Non-Isomorphic Embeddings with Iterative Normalization. Mozhi Zhang, Keyulu Xu, Ken-ichi Kawarabayashi, Stefanie Jegelka, and Jordan Boyd-Graber. ACL 2019.
> [2] Loss in translation: Learning bilingual word mapping with a retrieval criterion. Armand Joulin, Piotr Bojanowski, Tomas Mikolov, Herve J´egou, and Edouard Grave. EMNLP 2018.

---

> > ### Public Comment · ~Mozhi_Zhang1 · 2019-10-08
> > **Clarification and code**
> >
> > The RCSLS results in [1] and [2] are different because they use slightly different preprocessing steps. The results reported in [1] are sometimes higher than [2] (e.g., EN-FR, EN-ZH). Therefore, it may be helpful to report both [1] and [2] for a more complete picture of SOTA alignment methods.
> >
> > The code for [1] can be find here: https://gist.github.com/zhangmozhi/1e37c997514115e9b63476e322ca2ad0
> >
> > Hope these comments are helpful to your work

---

> > > ### Author Response · Authors · 2019-10-09
> > > **Thanks**
> > >
> > > Thank you for the constructive feedback. We agree and will certainly update Table 1 with numbers from [1] for a more complete picture.

---

### Public Comment · ~Ari_Sols1 · 2019-10-23
**Paper published elsewhere during the review period**

The work seems interesting.

Since I'm interested in the topic of bilingual/crosslingual embeddings, so I follow ICLR 2020 submissions as well as I receive Google Scholar alerts regularly about recently published papers. Around 10 days ago I received Google Scholar update with your paper listed (non-anonymous version), as it was uploaded to arXiv this October, so during the review period. How does it stand in terms of a double-blind policy?

---

> ### Author Response · Authors · 2019-10-23
> **Regarding submission to arxiv.**
>
> Thank you for the interest. Regarding the double blind policy and submission to arxiv, please see the page "https://iclr.cc/Conferences/2020/CallForPapers", where under the "Dual Submission Policy" section it states "Submission of the paper to archival repositories such as arXiv are allowed."

---

### Decision · Program_Chairs · 2019-12-19

**Decision:**

Accept (Poster)

**Comment:**

Reviewer worries include: whether the approach scales to distant language pairs, overselling of the paper as a "framework", a few citations and comparisons missing. I agree and encourage the authors not to use the word "framework" here. I would also encourage the authors to evaluate on more interesting language pairs, and analyze what vocabularies are relocated, as well as what their method is better at compared to previous work.